# Changes over a decade in psychotropic prescribing for people with intellectual disabilities: prospective cohort study

Angela Henderson ![ORCID],[1] Paula Mcskimming,[2] Deborah Kinnear,[1] Colin McCowan,[3] Alasdair McIntosh,[2] Linda Allan,[1] Sally-Ann Cooper ![ORCID] [1]

[1]Institute of Health and Wellbeing, University of Glasgow, Glasgow, UK
[2]Robertson Centre for Biostatistics, University of Glasgow, Glasgow, UK
[3]School of Medicine, University of Saint Andrews, Saint Andrews, UK

**Correspondence to**
Professor Sally-Ann Cooper;
Sally-Ann.Cooper@glasgow.ac.uk

## ABSTRACT

**Objectives** To investigate psychotropic prescribing in the intellectual disabilities population over 10 years, and associated mental ill health diagnoses.

**Design** Comparison of cross-sectional data in 2002–2004 (T1) and 2014 (T2). Longitudinal cohort study with detailed health assessments at T1 and record linkage to T2 prescribing data.

**Setting** General community.

**Participants** 1190 adults with intellectual disabilities in T1 compared with 3906 adults with intellectual disabilities in T2. 545/1190 adults with intellectual disabilities in T1 were alive and their records linked to T2 prescribing data.

**Main outcome measures** Encashed regular and as-required psychotropic prescriptions.

**Results** 50.7% (603/1190) of adults in T1 and 48.2% (1881/3906) in T2 were prescribed at least one psychotropic; antipsychotics: 24.5% (292/1190) in T1 and 16.7% (653/3906) in T2; antidepressants: 11.2% (133/1190) in T1 and 19.1% (746/3906) in T2. 21.2% (62/292) prescribed antipsychotics in T1 had psychosis or bipolar disorder, 33.2% (97/292) had no mental ill health or problem behaviours, 20.6% (60/292) had problem behaviours but no psychosis or bipolar disorder. Psychotropics increased from 47.0% (256/545) in T1 to 57.8% (315/545) in T2 (p<0.001): antipsychotics did not change (OR 1.18; 95% CI 0.87 to 1.60; p=0.280), there was an increase for antidepressants (OR 2.80; 95% CI 1.96 to 4.00; p<0.001), hypnotics/anxiolytics (OR 2.19; 95% CI 1.34 to 3.61; p=0.002), and antiepileptics (OR 1.40; 95% CI 1.06 to 1.84; p=0.017). Antipsychotic prescribing increased for people with problem behaviours in T1 (OR 6.45; 95% CI 4.41 to 9.45; p<0.001), more so than for people with other mental ill health in T1 (OR 4.11; 95% CI 2.76 to 6.11; p<0.001).

**Conclusions** Despite concerns about antipsychotic prescribing and guidelines recommending their withdrawal, it appears that while fewer antipsychotic prescriptions were initiated by T2 than in T1, people were not withdrawn from them once commenced. People with problem behaviours had increased prescribing. There was also a striking increase in antidepressant prescriptions. Adults with intellectual disabilities need frequent and careful medication reviews.

## INTRODUCTION

Mental ill health is common in people with intellectual disabilities.[1] The prevalence of

### Strengths and limitations of this study

► The large cohort size, longitudinal design, detailed ascertainment of the population with intellectual disabilities, and the in-depth health assessments at T1.

► The cross-sectional cohorts were population based at T1 and T2, and representative of the population with intellectual disabilities; the linked cohort had similar characteristics to the cross-sectional cohort at T1, suggesting this cohort is also representative and therefore that the results are generalisable.

► Only 73% of general practices agreed to data extraction, and this combined with deaths are likely to be the main reasons for 545/1190 of the participants being linked in the T2 data 10 years later.

► The different methods of data collection, with specialist individual assessments at T1 and electronic data extraction at T2; in particular, a large proportion of information is missing and inaccuracies might exist relating to recorded level of intellectual disabilities in the general practitioner data at T2, limiting comparability of this variable between the T1 and T2 cohorts.

► The study did not investigate changes in dosages, polypharmacy or duration of use, and mental ill health data at T2 are lacking.

psychosis in this population is reported to be around 4% based on cross-sectional data, and the rate of people with a first psychotic episode is about 10 times that of the general population.[2] While the rates of psychosis are relatively high, antipsychotics are often prescribed for adults with intellectual disabilities who do not have a record of severe mental ill health,[3 4] often for problem behaviours,[5–9] and despite limited evidence to support their use beyond short-term sedation.[7] Indeed, 71% of people with intellectual disabilities who are prescribed antipsychotics have been reported to have no record of serious mental ill health.[10] This is important because antipsychotics have numerous disabling, painful and disfiguring side effects, some of which are life

threatening, such as tardive dyskinesia, cardiac arrhythmias and sudden cardiac death.[11–13] Antipsychotics are also frequently prescribed for children and young people with a range of developmental disabilities and problem behaviours,[14 15] and in the young general population, rates increase during adolescence.[16]

Concerns have repeatedly been raised about the overuse of antipsychotics, and the need for more proportionate prescribing for people with intellectual disabilities.[7 17–19] In 2016 a national campaign was launched by NHS England in partnership with the Royal Colleges of General Practitioners, Psychiatrists, and Nursing, the Royal Pharmaceutical Society, and the British Psychological Society to address these concerns in England: 'Stopping over medication of people with a learning disability, autism or both (STOMP)'. Guidelines from STOMP, the National Institute for Health and Care Excellence, and the Royal College of Psychiatrists highlight that prescribers, where appropriate, should reduce or withdraw antipsychotics for people with intellectual disabilities who do not have psychosis.[7 20 21] However, there is very little empirical evidence from the UK on any changes in antipsychotic prescribing patterns over time. An exception is a study by Sheehan and colleagues that extracted data from general practice records on 33 016 adults with a record of intellectual disabilities, with a median follow-up of 5.5 years.[10] These authors reported the incidence of new psychotropic prescription to be 518/10 000 person years. Prescriptions of antipsychotics fell by 4% per year over the study period, as did mood stabilisers, while there was no consistent trend for antidepressants or anxiolytics/hypnotics. Sheehan and colleagues reported that 47% of those with 'challenging behaviour' had received antipsychotic drugs, but only 12% had a record of severe mental ill health, and that 26% of those prescribed antipsychotics did not have a record of severe mental ill health or 'challenging behaviour'. A limitation of this study is in the identification of 'challenging behaviour' through a heterogeneous list of 45 Read codes (the system used in general practices in the UK to code diagnoses). Read codes do not provide a robust method for ascertaining problem behaviours. Additionally, incomplete and variable recording practices do not always accurately reflect a person's health.[10]

Another study from Australia investigated psychotropic medication use between 1999 and 2015 in a cohort of 138 participants[22] and also found a strong association between problem behaviours and psychotropic medication. In this cohort the study reported that once psychotropic medications were prescribed they were unlikely to be removed, and little change was observed in prescribing of antipsychotics between 1999 and 2015 (24/138 (24%) to 23/92 (23%)). A sharp increase in the prescribing of antidepressants from 16.7% to 36.1% across the same period was also observed. However, while this was a longitudinal cohort, not all participants took part in all waves of data collection, therefore it is not possible to ascertain within group changes in prescribing.

## Aim

The aim of this study was to investigate changes over a decade in psychotropic prescribing for adults with intellectual disabilities, and the diagnoses associated with antipsychotics by using detailed psychiatric assessments.

## METHODS
### Ethical approval

Between 2002 and 2004 (T1), individual consent to participate was taken in line with Scottish law. In 2014 (T2), 191/263 (73%) general practices in NHS Greater Glasgow and Clyde area participated, and the NHS Greater Glasgow and Clyde Local Privacy Advisory Committee approved electronic extraction and linkage of primary care records.

### Participants

In 2000–2001, a primary care intellectual disabilities register was established of adults with intellectual disabilities, aged 16 years and older, living in the NHS Greater Glasgow area. This initiative was delivered through partnership between the intellectual disabilities clinical service and all general practitioners in the area. People with intellectual disabilities were identified through social work services for people with intellectual disabilities; local authority funding arrangements for people receiving paid support of any kind, including day opportunities; local specialist health services for people with intellectual disabilities; the Health Board; and general practices who were financially incentivised to identify their registered patients with intellectual disabilities (100% of general practices participated). Intellectual disabilities nurses reviewed all cases on the register to determine if intellectual disabilities were present; those that did not have intellectual disabilities were removed from the register. The register was then updated annually by the general practices and the intellectual disabilities clinical service.

Between 2002 and 2004, the register was used to invite people living in a representative part of the Health Board area to participate in the study; 67% agreed to take part. These participants were recruited to a longitudinal cohort between 2002 and 2004 (T1), and had detailed health assessments at that time; 1190 were aged 18 years and older and comprise the study population reported here. In 2014 (T2), for people on the register and living in the Health Board area, data were extracted from primary care records; 73% of general practices in the Health Board area agreed to the data extraction. Data were extracted on 3906 patients with intellectual disabilities aged 18 years and older, who comprise the study population reported here.

### Process and measures

Semi-structured individual health assessments, including medication review, assessment of level of intellectual disabilities (via structured questions on abilities, and the Vineland Scale[23]), mental ill health symptoms including

problem behaviours and autism, were conducted at T1 by one of six intellectual disabilities nurses and one of three general practitioners with a special interest in intellectual disabilities. These assessments were preceded by data collection from the person's general practitioner medical records, and then a review conducted with the person with intellectual disabilities and their carer(s). This included a review of drug charts for participants in supported care. The 54% of individuals identified with possible, probable or definite mental ill health (including problem behaviours and autism) were then assessed by the study psychiatrists who were specialists in intellectual disabilities psychiatry. Information from each person's psychiatric assessment was reviewed by two psychiatrists who had a case conference and agreed the classification of the mental ill health using ICD-10-DCR,[24] DSM-IV-TR,[25] DC-LD[26] and clinical criteria. Details have been previously reported.[1] Given that ICD-10 and DSM criteria function poorly for adults with moderate to severe intellectual disabilities (particularly with regards to problem behaviours), in this study we report the clinical diagnoses agreed by the study psychiatrists. Data collection was over a 2-year period. Drugs were coded using British National Formulary (BNF) codes.

At T2, the 3906 adults with intellectual disabilities identified from primary care records were record linked to Prescribing Information System (PIS) data by using the Community Health Index (CHI), which is the NHS patient identification number, unique to each person. PIS is Scotland's electronic record of all encashed prescriptions (ie, not prescriptions issued, or drugs administered, but those that the carers/person with intellectual disabilities actually took to a pharmacist and exchanged for the drugs), and includes a record of the BNF code of each prescribed drug.[27] Prescribing information was then extracted using BNF codes for the 3906 adults with intellectual disabilities to identify all prescriptions of antipsychotics, antidepressants, antiepileptics, lithium, and hypnotics/anxiolytics across a specific 12-week prescribing window in 2014, including both regular prescriptions and as-required medication. To establish the longitudinal cohort the CHI number was used to identify T1 participants in the T2 dataset, enabling comparison of encashed medications across the decade. Only participants with complete data who were aged 18 years and older were included in the analyses (figure 1).

### Statistical analysis

Subject characteristics and prescribing information were summarised descriptively with mean and SD for continuous outcomes and number and percentage for categorical outcomes at each time point (T1 and T2). Prescribing information at each time point was summarised using binary variables for each class of medications of interest (yes/no), allowing prescribing patterns to be investigated between the two time points in the study using McNemar's tests on the subset of the linked cohort, for whom there were prescribing records at both T1 and

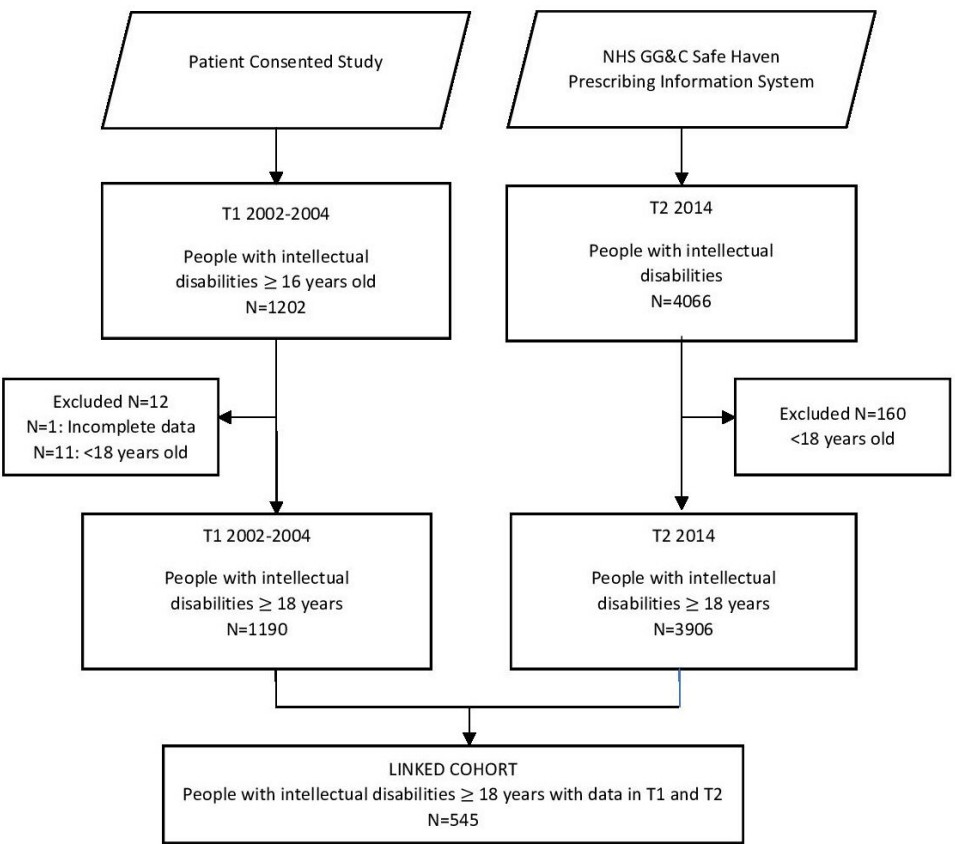

**Figure 1** Participant flow diagram. GG&C, Greater Glasgow and Clyde.

T2. This analysis was extended to explore whether there were associations between time or the subject characteristics at T1, with each prescription category using binary logistic regression models. Each model included multiple explanatory variables; specifically, time as a binary variable to indicate each time point T1 and T2; sex; age as a continuous measure; level of intellectual disabilities as four-level categorical variable; presence of mental ill health (yes/no, excluding problem behaviours); having problem behaviours (yes/no); and a binary dependant variable for each class of medication (yes/no). Logistic regression models were also fitted with the above T1 subject characteristics to explore their association with each prescribing category specifically at T2. Odds ratios are reported for all logistic regression models with corresponding 95% confidence intervals (CIs) and p values. A p value less than 0.05 is considered statistically significant. Statistical analyses were conducted using SAS version 9.3.

### Patient and public involvement
The Scottish Learning Disabilities Observatory has a steering committee which meets twice a year and provides strategic direction and oversight of all of the Observatory's research, including this project. The steering committee includes people with intellectual disabilities from 'People First', a national group of self-advocates with intellectual disabilities.

### RESULTS
### Participant characteristics of the cross-sectional cohorts
Data for those who had incomplete data at T1 and for those who were aged under 18 years at either time point were excluded from further analyses. Table 1 shows participant characteristics: age, sex, level of intellectual disabilities at T1 (n=1190) and T2 (n=3906), and mental ill health and epilepsy diagnoses at T1. No mental health or epilepsy data were available at T2.

### Prescribing for the cross-sectional cohorts
At least one psychotropic was prescribed at T1 for 50.7% (603/1190) and at T2 for 48.2% (1881/3906) (table 2) of participants. Antipsychotics were prescribed to 24.5% (292/1190) of participants at T1 and 16.7% (653/3906) at T2. At T1, antidepressants were prescribed for 11.2% (133/1190) and at T2 for 19.1% (746/3906) of participants. Hypnotic/anxiolytic, lithium and anti-epileptic prescribing was similar at T1 and T2.

The types of mental ill health experienced by the 292 participants at T1 who were taking antipsychotics are shown in table 3. The most common diagnosis within this group was problem behaviours at 40.8% (119/292). Of note, 33.2% (97/292) of the people taking antipsychotics did not have any identified mental ill health or problem behaviours. Figure 2 demonstrates the overlap between groups of the people who were taking antipsychotics at T1 and selected diagnoses.

**Table 1** Participant characteristics for the cross-sectional cohorts at T1 and T2

| Characteristic | T1 aged ≥18 years (n=1190) | T2 aged ≥18 years (n=3906) |
|---|---|---|
| Age, mean (SD) | 44.6 (14.3) | 45.4 (15.5) |
| Sex, No (%) | | |
| Male | 671 (56.4) | 2260 (57.9) |
| Female | 519 (43.6) | 1646 (42.1) |
| Level of intellectual disabilities, No (%) | | |
| Mild | 451 (37.9) | 1047 (26.8) |
| Moderate | 319 (26.8) | 859 (22.0) |
| Severe | 233 (19.6) | 595 (15.2) |
| Profound | 187 (15.7) | 197 (5.0) |
| Unknown | 0 | 1208 (30.9) |
| Epilepsy, No (%) | 419 (35.2%) | Not collected |
| Type of mental ill health, No (%) | | |
| Psychosis, including psychosis in remission | 52 (4.4) | Not collected |
| Problem behaviours | 244 (20.5) | |
| Autism | 80 (6.7) | |
| Attention deficit hyperactivity disorder | 15 (1.3) | |
| Unipolar depression | 51 (4.3) | |
| Bipolar disorder | 21 (1.8) | |
| Anxiety disorders | 32 (2.7) | |
| Organic disorder | 20 (1.7) | |
| Personality disorder | 9 (0.8) | |
| Obsessive compulsive disorder | 7 (0.6) | |
| Psychosexual disorder | <5 | |
| Other | 15 (1.3) | |
| Mental ill health (including problem behaviours) | 438 (36.8) | |
| Mental ill health (excluding problem behaviours) | 194 (16.3) | |

Table 3 also shows the types of mental ill health experienced by the 230 participants at T1 who were taking antipsychotics, after excluding people with psychosis (or psychosis in remission) or bipolar disorder (given that they would be expected to be prescribed antipsychotics, and given the considerable overlap between disorders shown in figure 2). Most strikingly, 97/230 (42.2%) of those prescribed antipsychotics had no mental ill health or problem behaviours. The proportion of people in each diagnostic category, without co-occurring psychosis or bipolar disorder who were taking antipsychotics was considerable for all types of mental ill health: 11.7% (27/230) for autism, 7.0% (16/230) for unipolar depression, 2.6% (6/230) for anxiety disorders and 2.2% (5/230) or less for all other diagnoses.

**Table 2** Psychotropics prescribed for the cross-sectional cohorts at T1 and T2

| Prescriptions | T1 aged ≥18 years (n=1190) | T2 aged ≥18 years (n=3906) |
|---|---|---|
| Any psychotropic drug | 603 (50.7) | 1881 (48.2) |
| Antipsychotics | 292 (24.5) | 653 (16.7) |
| Antidepressants | 133 (11.2) | 746 (19.1) |
| Antiepileptics | 333 (28.0) | 1028 (26.3) |
| Lithium | 14 (1.2) | 31 (0.8) |
| Hypnotics/anxiolytics | 81 (6.8) | 305 (7.8) |
| Missing data | 0 | 3 (0.1) |

## Participant characteristics of the longitudinal, linked cohort

The longitudinal, linked cohort included the 545 adults who were in the T1 cohort and who were also identified within the GP records at T2. Table 4 shows their age, sex, level of intellectual disabilities, epilepsy diagnosis and mental ill health at T1. They appear to be broadly representative of the whole cohort at T1 based on these characteristics.

**Table 3** Types of mental ill health at T1 experienced by people prescribed antipsychotics at T1, and after excluding people with psychosis and bipolar disorder

| Mental ill health at T1 | | Adults (≥18 years) taking antipsychotics at T1 | |
|---|---|---|---|
| Diagnosis | All (n=1190) | All (n=292) | Excluding people with psychosis and bipolar disorder (n=230) |
| Psychosis, including psychosis in remission | 52 | 45 (15.4) | – |
| Problem behaviours | 244 | 119 (40.8) | 100 (43.5) |
| Autism | 80 | 30 (10.3) | 27 (11.7) |
| Attention deficit hyperactivity disorder | 15 | 12 (4.1) | 11 (4.8) |
| Unipolar depression | 51 | 20 (6.9) | 16 (7.0) |
| Bipolar disorder | 21 | 17 (5.8) | – |
| Anxiety disorders | 32 | 7 (2.4) | 6 (2.6) |
| Organic disorder | 20 | 5 (1.7) | <5 |
| Personality disorder | 9 | 5 (1.7) | <5 |
| Obsessive compulsive disorder | 7 | 3 (1.0) | <5 |
| Psychosexual disorder | <5 | <5 | <5 |
| Other | 15 | 6 (2.1) | 5 (2.2) |
| Mental ill health (including problem behaviours) | 438 | 195 (66.8) | 133 (57.8) |
| Mental ill health (excluding problem behaviours) | 194 | 76 (26.3) | 33 (14.4) |
| No mental ill health or problem behaviours | 752 | 97 (33.2) | 97 (42.2) |

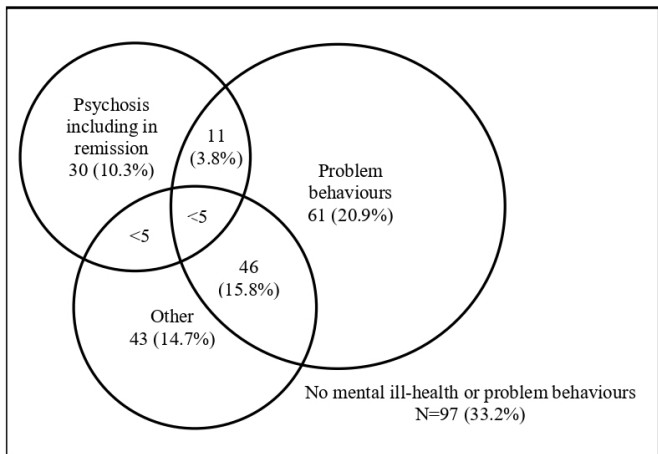

**Figure 2** Types of mental ill health experienced by people prescribed antipsychotics at T1 (n=292)

## Prescribing for the longitudinal, linked cohort

At least one psychotropic medication was prescribed for 47.0% (256/545) at T1 and for 57.8% (315/545) at

**Table 4** Participant characteristics at T1 for people in the longitudinal cohort

| Characteristic | T1 (aged ≥18 years) n=545 (%) |
|---|---|
| Age, mean (SD) | 41.8 (13.2) |
| Sex, No (%) | |
| Male | 322 (59.1) |
| Female | 223 (40.9) |
| Level of intellectual disabilities, No (%) | |
| Mild | 237 (43.5) |
| Moderate | 154 (28.3) |
| Severe | 89 (16.3) |
| Profound | 65 (11.9) |
| Epilepsy, No (%) | 173 (31.7) |
| Type of mental ill health, No (%) | |
| Psychosis, including psychosis in remission | 32 (5.9) |
| Problem behaviours | 109 (20.0) |
| Autism | 38 (7.0) |
| Attention deficit hyperactivity disorder | 8 (1.5) |
| Unipolar depression | 23 (4.2) |
| Bipolar disorder | 10 (1.8) |
| Anxiety disorders | 14 (2.6) |
| Organic disorder | <5 |
| Personality disorder | 7 (1.3) |
| Obsessive compulsive disorder | <5 |
| Psychosexual disorder | <5 |
| Other | 6 (1.1) |
| Mental ill health (including problem behaviours) | 190 (34.9) |
| Mental ill health (excluding problem behaviours) | 81 (14.9) |

**Table 5** Psychotropic medications prescribed for the longitudinal, linked cohort at T1 and T2

| Medication group | T1 (n=545) | T2 (n=545) | p-value |
|---|---|---|---|
| Any psychotropic medication | 256 (47.0%) | 315 (57.8%) | p<0.001 |
| Antipsychotics | 128 (23.5%) | 142 (26.1%) | p=0.099 |
| Antidepressants | 54 (9.9%) | 120 (22.0%) | p<0.001 |
| Hypnotics/anxiolytics | 25 (4.6%) | 51 (9.4%) | p<0.001 |
| Antiepileptics | 135 (24.8%) | 169 (31.0%) | p<0.001 |
| Lithium | 7 (1.3%) | 10 (1.8%) | p=0.180 |

T2 (table 5), which is a significant increase over time (p<0.001). Antidepressants were prescribed for 9.9% (54/545) of participants at T1 and for 22.0% (120/545) at T2, showing a significant increase (p<0.001). At T1, hypnotics/anxiolytics were prescribed for 4.6% (25/545) of participants, and at T2 for 9.4% (51/545), a significant increase (p<0.001). At T1, antiepileptics were prescribed for 24.8% (135/545) of participants, and at T2 for 31.0% (169/545), a significant increase (p<0.001). Prescribing patterns at T1 and T2 were similar for both antipsychotics and lithium. Of the 128 people prescribed antipsychotics at T1, 77.3% (99/128) were prescribed antipsychotics at T2; only 29 (22.7%) had been withdrawn, and 43/545 (7.9%) had started on an antipsychotic between the two timepoints.

The logistic regression analyses, taking account of sex, age, level of intellectual disabilities, having mental ill health (excluding problem behaviours) and having problem behaviours at T1 (table 6) show no significant difference in antipsychotic prescribing rate in the linked cohort over the decade (OR=1.18; 95% CI 0.87 to 1.60; p=0.280), an increase in antidepressants (OR 2.80; 95% CI 1.96 to 4.00; p<0.001), hypnotics/anxiolytics (OR 2.19; 95% CI 1.34 to 3.6; p=0.002), and antiepileptic prescribing (OR 1.40; 95% CI 1.06 to 1.84; p=0.017). Sex was not independently associated with prescribing, except that women were more likely to have an increase in antidepressants than men after adjusting for time (OR 0.53; 95% CI 0.37 to 0.78; p<0.001). Older age had a small effect on prescribing for antipsychotics and antidepressants.

Effects are also observed for level of intellectual disabilities. There was a gradient for antiepileptics (increased prescribing with increasing severity of intellectual disabilities) and a gradient for antidepressants (reduced prescribing with increasing severity of intellectual disabilities). However, there was no gradient across different ability levels for antipsychotic prescribing. As expected, participants with a diagnosed mental health problem (excluding problem behaviours) at T1 were more likely to be prescribed antipsychotics (OR 4.11; 95% CI 2.76 to 6.11; p<0.001), antidepressants (OR 3.90; 95% CI 2.53 to 6.02; p<0.001), and hypnotics/anxiolytics (OR 3.25; 95% CI 1.78 to 5.94; p<0.001). Strikingly though, people with problem behaviours identified at T1 were over six

**Table 6** Multivariable analysis of exploratory T1 factors and time with psychotropic prescriptions for the linked cohort (n=545)

| | Antipsychotics | | Antidepressants | | Hypnotics/anxiolytics | | Antiepileptics | | Lithium* | |
|---|---|---|---|---|---|---|---|---|---|---|
| | OR (95% CI) | p-value | OR (95% CI) | p-value | OR (95% CI) | p-value | OR (95% CI) | p-value | OR (95% CI) | p-value |
| Time | 1.18 (0.87 to 1.60) | 0.280 | 2.80 (1.96 to 4.00) | <0.001 | 2.19 (1.34 to 3.60) | 0.002 | 1.40 (1.06 to 1.84) | 0.017 | 0.69 (0.26 to 1.84) | 0.4612 |
| Male sex | 0.99 (0.73 to 1.34) | 0.954 | 0.53 (0.37 to 0.76) | <0.001 | 0.83 (0.52 to 1.35) | 0.456 | 1.02 (0.77 to 1.35) | 0.896 | 0.93 (0.34 to 2.59) | 0.890 |
| Age at T1 | 1.04 (1.03 to 1.05) | <0.001 | 1.02 (1.00 to 1.03) | 0.010 | 1.01 (0.99 to 1.03) | 0.260 | 0.99 (0.98 to 1.00) | 0.057 | 0.96 (0.93 to 0.99) | 0.016 |
| Level of intellectual disabilities (compared with mild intellectual disabilities) | – | <0.001 | – | 0.001 | – | 0.444 | – | <0.001 | – | 0.094 |
| Moderate | 1.88 (1.30 to 2.74) | <0.001 | 0.82 (0.54 to 1.24) | 0.346 | 1.64 (0.92 to 2.92) | 0.093 | 1.78 (1.26 to 2.51) | <0.001 | 0.23 (0.07 to 0.71) | 0.011 |
| Severe | 2.49 (1.61 to 3.85) | <0.001 | 0.62 (0.37 to 1.03) | 0.063 | 1.12 (0.55 to 2.31) | 0.750 | 2.30 (1.55 to 3.41) | <0.001 | 1.49 (0.17 to 13.36) | 0.721 |
| Profound | 1.61 (0.97 to 2.69) | 0.067 | 0.22 (0.11 to 0.46) | <0.001 | 1.24 (0.58 to 2.62) | 0.579 | 4.73 (3.07 to 7.31) | <0.001 | 0.46 (0.08 to 2.64) | 0.386 |
| Mental ill health† | 4.11 (2.76 to 6.11) | <0.001 | 3.90 (2.53 to 6.02) | <0.001 | 3.25 (1.78 to 5.94) | <0.001 | 1.13 (0.76 to 1.70) | 0.547 | – | – |
| Problem behaviours | 6.45 (4.41 to 9.45) | <0.001 | 3.44 (2.22 to 5.35) | <0.001 | 3.06 (1.72 to 5.44) | <0.001 | 1.27 (0.90 to 1.81) | 0.174 | – | – |

*Mental illness and problem behaviours excluded from lithium model due to small numbers.
†Not including problem behaviours.

times more likely to have increased prescribing of an antipsychotic (OR 6.45; 95% CI 4.41 to 9.45; p<0.001), over three times more likely for antidepressants (OR 3.44; 95% CI 2.22 to 5.35; p<0.001) and three times more likely for hypnotics/anxiolytics (OR 3.06; 95% CI 1.72 to 5.44; p<0.001).

The further regression (online supplementary table 1) investigating factors at T1 which are associated with prescribing at T2 (as opposed to change in prescribing reported in the paragraph above) shows that women were more likely to be prescribed antidepressants at T2, that older age had a small effect for antipsychotics and antidepressants at T2, a gradient across ability level for antiepileptics, a relationship with moderate and severe (but not profound) intellectual disabilities for antipsychotics at T2, and fewer antidepressants for people with profound intellectual disabilities. Mental ill health and problem behaviours at T1 predicted prescribing of all classes.

## DISCUSSION
### Principal findings
Despite numerous calls and guidelines in the UK for the withdrawal of antipsychotic drugs from people with intellectual disabilities who do not have psychosis/bipolar disorders,[7 20 21] our longitudinal, linked cohort analysis shows no progress over a decade. The comparison of the two cross-sectional cohorts does show a lower rate of antipsychotic prescribing in T2 than was observed in T1, but the rate is still high in T2 at 16.7% of the population. It appears that while few people are being withdrawn from antipsychotics once they start them, new antipsychotic prescriptions are less commonly initiated than in the past. Over the decade, comparison of both the cross-sectional cohorts, and of the longitudinal, linked cohort, reveal a striking increase in the prescription of antidepressants (11.2% to 19.1%, and 9.9% to 22.0%). This was particularly so for women and for people with mild intellectual disabilities. To a lesser extent, there were also increases in prescribing of hypnotics/anxiolytics and antiepileptics in the linked cohort, but not in the comparison of the cross-sectional cohorts. This difference may be accounted for by the known increase in these prescriptions with age,[5] as the linked cohort is of course 10 years older in T2, whereas age and sex are similar in the whole cohorts in T2 and in T1. The age-related change in antiepileptic prescribing in the longitudinal linked cohort, but not in the comparison of the similarly aged cross-sectional cohorts, contextualises the antipsychotic and antidepressant findings (prescribing trends in general) because antiepileptics were almost all prescribed for the highly prevalent condition of epilepsy in this population. While previous studies have reported high rates of antipsychotic prescribing, we are not aware of any that have investigated prescribing over this length of time along with related fluctuations in assessed mental ill health.

### Comparison with previous literature
To our knowledge only two studies have investigated longitudinal psychotropic prescribing patterns in community-based samples of people with intellectual disabilities in the UK. Both studies were large and relied on data extracted from primary care records. One reported antipsychotic prescribing for 17.1%, and antidepressants for 16.9% of adults with intellectual disabilities, with age being associated with both, and sex with antidepressants, similar to our T2 results.[5] The results of the other study differed, reporting antipsychotic prescribing in 27.7% of participants at the end of their study period, but also reporting a decrease of 4% per year over the whole study period, and no consistent trend in antidepressant prescriptions was reported.[5] Neither study conducted psychiatric assessments on the population, limiting the precision of findings related to clinical diagnosis and GP-recorded symptoms.

This study reaffirms the strong association between antipsychotic prescribing and problem behaviours reported in a number of other studies.[28–31] However, few studies have separately reported associations between problem behaviours and antidepressants or hynotics/anxiolytics. An Irish study which investigated rates of prescribing of psychotropics in older adults with intellectual disabilities reported no increased risk of antidepressant prescribing or any association with problem behaviours.[32]

Several studies have reported the increase in rates of antidepressant prescribing in the general population across the UK, which our findings mirror.[33–35] In Scotland the number of antidepressant prescriptions rose from 1.16 to 3.53 million per year between 1992 and 2006,[36] and women were prescribed antidepressants more frequently than men.[33] The reasons for the increase are unclear and have been attributed to multiple factors such as the availability of newer classes of drugs with fewer side effects, improved management of depression, lack of availability of alternative interventions,[36] a widening of clinical uses[33] and patient expectations. Earlier studies have cited concerns that depression may have been underdiagnosed in the population with intellectual disabilities.[37] One American study which retrospectively analysed outpatient psychiatric charts reported a higher than expected rate of antidepressant prescribing for the subgroup with intellectual disabilities and suggested this was indicative of increasing diagnosis of depressive disorders in adults with intellectual disabilities.[38] Another US study analysed data from adults with intellectual disabilities living in community settings in New York State between 2006 and 2007 and also reported a higher than expected rate of antidepressant prescribing in this group.[39] The substantial increase in antidepressant prescribing observed in the current study may indicate improved diagnosis in primary care for this population.[24] This study has also observed that problem behaviours were independently associated with antidepressant prescribing in adults with intellectual disabilities. However a systematic review of antidepressants and problem behaviour management in people

with intellectual disabilities concluded that evidence of their effectiveness in this context is lacking.[40] Longitudinal patterns of antidepressant prescribing require further investigation.

### Strengths and limitations

Strengths of the study include its large size, the longitudinal design, the detailed ascertainment of the population with intellectual disabilities, and the detailed health assessments at T1. The cross-sectional cohorts were population based at T1 and T2, and so were more widely representative of the population with intellectual disabilities. Additionally, the linked cohort was similar in characteristics to the whole cohort at T1, suggesting it is representative and hence that the results are generalisable. The 12-week extraction period of PIS data was selected to account for the frequency of prescriptions being issued; it included both regular and as-required drugs. Given the 12-week prescribing period it is likely that the as-required drugs were being actively used (as a fresh prescription had been issued and was encashed by the person with intellectual disabilities/their carer during this period). The time period for encashment was identical at both time points for the longitudinal, linked cohort. As a matter of caution in interpreting the data, the case-conferenced clinical mental ill health diagnoses agreed by the study psychiatrists were used rather than ICD-10 or DSM-IV-TR diagnoses, in view of the under-recording of mental ill health that these two classification systems produce with this population; had we used either of these classifications, our results would have been even more striking in terms of the discrepancy between mental ill health and prescription of antipsychotics.

Only 73% of general practices agreed to data extraction, and this combined with deaths are likely to be the main reasons for 545/1190 of the participants being linked in the T2 data 10 years later. Limitations are the different methods of data collection, with specialist individual assessments at T1 and electronic data extraction at T2. In particular, a large proportion of information is missing and there may be inaccuracies on the recorded level of intellectual disabilities in the general practitioner data at T2, so comparison of this variable between the T1 and T2 cohorts is limited. Additionally, mental ill health data at T2 are lacking. The study did not investigate changes in dosages, polypharmacy or duration of use. Some antipsychotic drugs are licenced for indications other than psychosis, and it is possible that other conditions accounted for their use, for example promazine. Antidepressants and antiepileptics have also seen increased use in the general population over this time period for neuralgic pain. We do not know how relevant this is to people with intellectual disabilities who may have difficulties in communicating pain, and note that encashed antiepileptics did not increase between the two cohorts.

### Implications for research and practice

This study has shown that fewer new antipsychotic prescriptions are being initiated, but patients prescribed antipsychotics in T1 were unlikely to have these drugs withdrawn over the next decade. This implies possible reluctance of carers, families and individuals to stop medications, combined with a lack of evidence available to prescribers about direct cessation interventions.[22 41] The issue therefore remains far from addressed, and the risks of long-term health problems, death and impact on quality of life associated with long-term antipsychotic prescriptions still need further highlighting.[42] This study reinforces the need for frequent medication reviews for people with intellectual disabilities, alongside further research to investigate the long-term effects of antipsychotic medications on this population.[8] Further research to examine the barriers to antipsychotic drug reduction and to evaluate approaches to promoting reduction and withdrawal of antipsychotics for people with intellectual disabilities is needed. There is a dearth of evidence on antidepressant prescribing in the population with intellectual disabilities. The sharp increase in antidepressant prescribing observed in this study demands further research to understand the drivers for this practice. The association between increasing age and prescribing of antipsychotics and antidepressants also supports calls for research to investigate the implications of long-term psychotropic prescribing for older people with intellectual disabilities.[43]

**Acknowledgements** We are grateful to all the participants and their carers, to NHS Greater Glasgow learning disabilities primary care liaison team, to NHS Greater Glasgow and Clyde Safe Haven. This work uses data provided by patients and collected by the NHS as part of their care and support.

**Contributors** AH and PM analysed the data, jointly interpreted it. AH wrote the first draft of the manuscript. AH, S-AC, DK, CM, and LA jointly conceived the project, interpreted the data and contributed to the manuscript. AM contributed to additional analyses in response to reviewer comments. All authors approved the final version of the manuscript. S-AC is the study guarantor.

**Funding** The study was funded by the Greater Glasgow Health Board, the Scottish Government via the Scottish Learning Disabilities Observatory and the UK Medical Research Council (grant number: MC_PC_17217). The study sponsors and funders had no role in the collection, analysis, and interpretation of data; in the writing of the report; and in the decision to submit the article for publication. The researchers are independent from the funders.

**Competing interests** All authors have completed the ICMJE uniform disclosure form at www.icmje.org/coi_disclosure.pdf and declare: no support from any organisation for the submitted work; no financial relationships with any organisations that might have an interest in the submitted work in the previous 3 years; no other relationships or activities that could appear to have influenced the submitted work .

**Patient consent for publication** Not required.

**Ethics approval** NHS Greater Glasgow Primary Care Trust, Community & Mental Health Research Ethics Committee granted ethical approval (project No 01/44).

**Provenance and peer review** Not commissioned; externally peer reviewed.

**Data availability statement** No data are available.

and indication of whether changes were made. See: https://creativecommons.org/licenses/by/4.0/.

**ORCID iDs**
Angela Henderson http://orcid.org/0000-0002-6146-3477
Sally-Ann Cooper http://orcid.org/0000-0001-6054-7700

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
