## [Reviewer comments · BMJ Open]

ARTICLE DETAILS

TITLE (PROVISIONAL)	Changes over a decade in psychotropic prescribing for people with intellectual disabilities: prospective cohort study
AUTHORS	Henderson, Angela; Mcskimming, Paula; Kinnear, Deborah; McCowan, Colin; McIntosh, Alasdair; Allan, Linda; Cooper, Sally-Ann

VERSION 1 – REVIEW

REVIEWER	Verdoux Helene University of Bordeaux, France
REVIEW RETURNED	13-Jan-2020

GENERAL COMMENTS	Investigating the frequency of psychotropic drugs prescription in persons with intellectual disability is of great interest considering that these persons are often exposed to off-label prescriptions, especially of antipsychotics. Unfortunately, the present study does not shed a new light on this issue due to major methodological drawbacks. In the introduction, the authors have to specify when was launched the STOMP campaign, as well as when was performed the study by Seehan et al. The link between the STOMP campaign and the present study is unclear, is this study aimed at assessing the impact of this campaign? The authors state that their study is population-based. However, this point cannot be verified in the method section, as no information is available regarding the fact that all persons fulfilling the inclusion criteria were identified in the general population at T1. Were all persons with disability in the general population identified at T1 using the multiple sources of information? If not, how is it possible to exclude a selection bias potentially impacting the representativity of the sample ? Indeed, the huge difference between 1200 persons included at T1 and 3906 persons identified at T2 is puzzling and should be further explained. The residential status (and change in this status over the follow-up) of persons with intellectual disability has to be specified as it may have a marked impact on medication. Multivariate models have to be adjusted for this characteristic. The same comment applies to psychiatric admission over the follow-up. Read coding has to be explained for non UK readers. Intellectual disability is a very broad diagnostic category: which diagnostic criteria were used by the intellectual disabilities primary
---

	care liaison team? How were defined levels of intellectual disabilities? Information on medication at T1 is poor. Considering that persons with intellectual disability may not be accurate sources for information on medication, which sources were used to collect this information and over which period of time? This is a major drawback as the method used to collect information is not the same at T1 and T2. Information on antipsychotics is of little interest if first-generation antipsychotics are not distinguished from second-generation antipsychotics. Does the category antiepileptics only include drugs licensed for mood disorders? If not, this category is also of little interest in a study exploring psychotropic prescription considering the prevalence of epilepsy in persons with intellectual disability. The classification of psychiatric disorders is not acceptable. Only ICD-10 or DSM diagnoses would be relevant.
--	---

REVIEWER	Gregory Peterson University of Tasmania, Australia
REVIEW RETURNED	18-Jan-2020

GENERAL COMMENTS	Thank you for the submission. The work is quite interesting. It should be very clearly throughout the manuscript (including abstract) whether the data refers to regularly administered drugs (and how that was defined), 'as required' (prn), or the total of both regular plus prn. This distinction is particularly important with antipsychotics and benzodiazepines. It is difficult to compare with published data otherwise. The method suggests that only prescriptions for a 12-week window were examined. It is unclear if regular vs. prn use could be determined. It should be acknowledged that the use of antidepressants (e.g. amitriptyline, duloxetine) and antiepileptics (e.g. pregabalin, gabapentin) for neuralgic pain has increased markedly over the timeframe examined (i.e. these drugs are often used for other conditions). Amitriptyline is a classic example that could have a multitude of uses in these patients (e.g. incontinence). Another limitation that should be included is the apparent lack of access to data on the individuals' other medical conditions (for which some of these drugs are commonly used...and more commonly than they were 10 years earlier). Please check the English of the second dot point under Strengths and limitations of the study. Under Discussion: heading should state "Principal findings"
--

REVIEWER	Koji Kawakami Kyoto University, Japan Dr. Kawakami reports other from Sumitomo Dainippon Pharma Co., Ltd., other from Stella Pharma Corporation, other from CMIC Co., Ltd., other from Suntory Beverage & Food Ltd., other from Kaken Pharmaceutical Co., Ltd., other from Astellas Pharma Inc., other from Mitsubishi Tanabe Pharma Co., other from AbbVie Inc., other from Santen Pharmaceutical Co., Ltd., other from Daiichi
-----------------	--

	Sankyo Co., Ltd., other from Takeda Pharmaceutical Co., Ltd., other from Boehringer Ingelheim Japan, Inc., other from School Health Record Centre Co., Ltd. , other from Real World Data Co., Ltd., outside the submitted work.
REVIEW RETURNED	21-Jan-2020

GENERAL COMMENTS	The paper by Henderson et al. entitled ‘Changes over a decade in psychotropic prescribing for people with intellectual disabilities: prospective cohort study’ is a study to investigate changes in psychotropic prescribing in the intellectual disabilities population over a 10 year period, and associated mental ill-health diagnoses. They found that the percentage of prescribed psychotropic medications in the late period was lower than in earlier periods. Also, they found people with problem behaviors had increased prescriptions of psychotropic drugs and antidepressants. It seems that the study was tired to well-consider, and the findings may be of interest to medical experts as well as public officials and researchers in this field. However, I have the following concerns. Major comments:  1. It would be helpful if the authors explain why they selected the two points; 2002-2004 and 2014. Is there any specific reason such as the guideline was changed 2013 or something? Also, is there any reason why they compared for three years and just one year? 2. It would be clearer if the authors explain the aims of this study. This study consisted of two designs. It seems they compared the percentage of prescribed psychotropic drugs in the whole cohort of patients with intellectual disabilities and evaluated change in prescribed psychotropic drugs among patients who could be followed up for 10 years among whole cohort. Did the authors try to evaluate the prescriptions were followed the guideline? If so, does the guideline recommend to withdraw the medication? I mean, the long term use of drugs is not recommended? 3. Patients' selection in each period is different. Is it reasonable to compare with the two-period s of groups directly? 4. It would be more informative if the authors explain why they selected ‘multivariable repeated measures logistic regression models’ for analysis. Also, the reference should be stated here because the model is not popular. 5. What is the definition of ‘change in prescription’. It should be stated in the statistical methods. 6. As for the longitudinal design, the results have an immortal bias. They have to be followed for ten years. How did the authors consider if the patients quitted the treatment before ten years? 7. As for the cross-sectional study, the prescriptions were not limited in newly prescribed drugs, right? 8. The authors mentioned, ‘It appears that whilst people are not being withdrawn from antipsychotics once they commence them’ in the discussion section. How we are able to know which prescriptions should be withdrawn. The patients might be needed the drugs use continuously. Minor comments;  1. In the abstract, the authors did not mention the result for longitudinal design. It should be stated. Also, the authors said ‘rates of prescribing,’ but it seems they did not consider the time such as person-year. Is ‘rates’ correct? I wonder it’s instead ‘percentage’ or ‘proportion.’ 2. Page 6, line 57. There is a typo: “reportedwith” to “reported with.”
--

REVIEWER	Ryan Smith Sultan Columbia University, USA
REVIEW RETURNED	23-Jan-2020

GENERAL COMMENTS	Potential overuse of psychotropic meds in vulnerable populations is of international interest. This article attempts to examine these trends in intellectually disability populations. Overall, writing is fairly casual and could benefit from a professional tone. Overall, Needs proof reading, last paragraph of process and measures “complete data” Overall, the article could benefit from a narrower focus. The article in some ways seem to be largely about antipsychotics—though the title says otherwise. Further, The data between T1 and T2 are different. Seems the article should be either about the comparison of the T 1 and T2 or looking at each individually. If it’s the comparison, id remove the stuff about diagnoses and rates of meds as it can’t be used in the second group (Table 3). Article would benefit from a clinical lens. For example, clear evidence exists that antipsychotic medications are over used in this population. However, the article does not adequately discuss how individuals with intellectual disabilities often demonstrate severe issues and become dysregulated and often physically aggressive. Low dosing of antipsychotic medications often moderate these issues for ID individuals and allow them to stay in housing situations and avoid longer term institutionalization. This data set isn’t able to examine that because these types of behavioral issues do not have a specific diagnostic code linked to them. Findings would benefit from a sub analysis that looked at level of intellectual disability (mild to profound) and likelihood of antipsychotic prescribing (and other psychotropics). I would imagine more profound ID is associated with higher prescribing and likely higher antipsychotic prescribing. Unsure of where the conclusion in the abstract about not withdrawing antipsychotics comes from. If that is a primary conclusion—it should be backed up by the results section of the abstract. If the goal of this article was to link it to the concerns over use of meds in this population, particularly antipsychotics, I think the abstract needs to be more positive and less concerning sounding as the numbers clearly show they are reducing Article would benefit from adding some citations from US data—which as examined this extensively, Some suggestions: Sultan, R. S., Wang, S., Crystal, S., & Olfson, M. (2019). Antipsychotic Treatment Among Youths With Attention-Deficit/Hyperactivity Disorder. JAMA Network Open, 2(7). Sultan, R. S., Correll, C. U., Schoenbaum, M., King, M., Walkup, J. T., & Olfson, M. (2018). National Patterns of Commonly Prescribed Psychotropic Medications to Young People. Journal of Child and Adolescent Psychopharmacology, 28(3), 158–165. https://doi.org/10.1089/cap.2017.0077
--

	Olfson, M., King, M., & Schoenbaum, M. (2015). Treatment of young people with antipsychotic medications in the United States. JAMA Psychiatry , 72(9), 867–874. https://doi.org/10.1001/jamapsychiatry.2015.0500
--	--

VERSION 1 – AUTHOR RESPONSE

Reviewer: 1

Verdoux Helene, University of Bordeaux, France

1. In the introduction, the authors have to specify when was launched the STOMP campaign, as well as when was performed the study by Sheehan et al. The link between the STOMP campaign and the present study is unclear, is this study aimed at assessing the impact of this campaign?

***Response: We have inserted the year of the study by Sheehan et al, and have amended the reference to the STOMP campaign to clarify that this is a campaign led by the NHS in England to address overprescribing of psychotropic medications to people with intellectual disabilities in England. The current study investigates psychotropic prescribing patterns in Scotland and is therefore not linked to STOMP.

The text has been amended as follows (page 4):

“In 2016 a national campaign was launched by NHS England in partnership with the Royal Colleges of General Practitioners, Psychiatrists, and Nursing, and the Royal Pharmaceutical Society, and British Psychological Society to address these concerns in England: “Stopping over medication of people with a learning disability, autism or both (STOMP)”.”

2. The authors state that their study is population-based. However, this point cannot be verified in the method section, as no information is available regarding the fact that all persons fulfilling the inclusion criteria were identified in the general population at T1. Were all persons with disability in the general population identified at T1 using the multiple sources of information? If not, how is it possible to exclude a selection bias potentially impacting the representativity of the sample ? Indeed, the huge difference between 1200 persons included at T1 and 3906 persons identified at T2 is puzzling and should be further explained.

***Response: We have expanded our description of the participants in the methods to better address this point. The case ascertainment has been outlined in detail. The difference in numbers is because at T1 the cohort was drawn from only part of the Health Board, whereas at T2 it covered the whole of the Health Board. We have added the following information (pages 5-6):

“In 2000-2001, a primary care intellectual disabilities register was established of adults with intellectual disabilities, aged ≥16 years, living in the NHS Greater Glasgow area through a joint process by the intellectual disabilities clinical service and all general practitioners in the area. People with intellectual disabilities were identified through social work services for people with intellectual disabilities; local authority funding arrangements for persons receiving paid support of any kind, including day opportunities; local specialist health services for people with intellectual disabilities; the health board; and general practices who were financially incentivised to identify their registered patients with intellectual disabilities (100% of general practices participated). Everyone so identified was checked that they had intellectual disabilities by intellectual disabilities nurses and those that did not were removed from the register. The register was then updated annually jointly by

the general practices and the intellectual disabilities service. Between 2002-2004, the register was used to invite people living in a *representative part of the Health Board area* to participate in the study, and 67% agreed to do so: these participants were recruited to a longitudinal cohort between 2002 and 2004 (T1), and had detailed health assessments at that time. 1,190 were aged ≥ 18 years, and comprise the study population reported here. In 2014 (T2), for people on the register and living in *the whole of the Health Board area*, data was extracted from primary care records; 73% of general practices in the Health Board agreed to the data extraction and so data was extracted on their 3,906 patients with intellectual disabilities aged ≥ 18 years, who comprise the study population reported here.”

3. The residential status (and change in this status over the follow-up) of persons with intellectual disability has to be specified as it may have a marked impact on medication. Multivariate models have to be adjusted for this characteristic.

The same comment applies to psychiatric admission over the follow-up.

***Response: We did not have access to this information and were; therefore, unable to incorporate this into our models. The study did not investigate psychiatric admissions. Only a small proportion of people with intellectual disabilities are admitted to psychiatric beds; for the whole of NHS Greater Glasgow, there were only 16 psychiatric assessment/treatment beds for people with intellectual disabilities.

4. Read coding has to be explained for non UK readers.

***Response: This has been explained as follows (page 4):

“.....Read codes (the system used in general practices in the UK to code diagnoses).”

5. Intellectual disability is a very broad diagnostic category: which diagnostic criteria were used by the intellectual disabilities primary care liaison team? How were defined levels of intellectual disabilities?

***Response: We have added the details (page 6)

“...assessment of level of intellectual disabilities (via structured questions on abilities, and the Vineland Scale (22)),.....”

6. Information on medication at T1 is poor. Considering that persons with intellectual disability may not be accurate sources for information on medication, which sources were used to collect this information and over which period of time? This is a major drawback as the method used to collect information is not the same at T1 and T2.

***Response: Information came from the person’s primary care medical records, and carers, in addition to the adults with intellectual disabilities, and also from drug charts for persons in supported care. We have provided additional information to address this in the methods (page 6):

“Semi-structured individual health assessments, including medication review, assessment of level of intellectual disabilities (via structured questions on abilities, and the Vineland Scale (22)), mental ill-health symptoms including problem behaviours and autism, were conducted at T1 by one of six intellectual disabilities nurses and one of three general practitioners with special interest in intellectual disabilities. These were preceded by review and data collection from the person’s

general practitioner medical records, and then conducted with the person with intellectual disabilities and their carer(s), and included review of drug charts for the individuals in supported care.”

7. Information on antipsychotics is of little interest if first-generation antipsychotics are not distinguished from second-generation antipsychotics.

***Response: We disagree with this point. Whilst there are differences in the side effect profiles of first and second generation antipsychotics, and indeed, between drugs within these two broad groupings, all these drugs are associated with a wide range of unpleasant side-effects, including some with neurological and cardio-toxic side effects that can cause death. Our paper demonstrates the wide discrepancy between psychiatric conditions that indicate the use of antipsychotics and the high level of prescription, and changes over time (less new prescriptions being commenced, but few being withdrawn), which is highly important for these individuals with intellectual disabilities.

8. Does the category antiepileptics only include drugs licensed for mood disorders? If not, this category is also of little interest in a study exploring psychotropic prescription considering the prevalence of epilepsy in persons with intellectual disability.

***Response: No, as stated in the text, it is all antiepileptic prescribing, regardless of indication. We have added the epilepsy prevalence into tables 1 and 4. We have also added the following in the discussion to expand our position on including antiepileptics (page 16):

“The age-related change in antiepileptic prescribing in the linked cohort, but not in the comparison of the similarly aged whole cohorts, contextualises the antipsychotic and antidepressant findings (prescribing trends in general), as antiepileptics were almost all prescribed for the highly prevalent condition of epilepsy in this population.”

9. The classification of psychiatric disorders is not acceptable. Only ICD-10 or DSM diagnoses would be relevant.

***Response: We realise that we did not explain this, and so now have added explanation in the methods as follows (page 6):

“Information from each person’s psychiatric assessment was reviewed by two psychiatrists who case-conferenced and agreed the classification of the mental ill-health, using ICD-10-DCR, DSM-IV-TR, DC-LD, and clinical criteria. Details have been previously reported. (1) Given that ICD-10 and DSM criteria function poorly for adults with intellectual disabilities (particularly with regards to problem behaviours), in this study we report the clinical diagnoses agreed together by the study psychiatrists.”

And also in the discussion (page 18): “As a matter of caution in interpreting the data, the case-conferenced clinical mental ill-health diagnoses agreed by the study psychiatrists were used rather than ICD-10 or DSM-IV-TR diagnoses, in view of the under-recording of mental ill-health that these two classificatory systems produce with this population: had we used either of these classifications, our results would have been even more striking in terms of the discrepancy between mental ill-health and prescription of antipsychotics.”

Reviewer 2

Gregory Peterson, University of Tasmania, Australia

1. It should be very clearly throughout the manuscript (including abstract) whether the data refers to regularly administered drugs (and how that was defined), 'as required' (prn), or the total of both regular plus prn. This distinction is particularly important with antipsychotics and benzodiazepines. It is difficult to compare with published data otherwise.

***Response: We have amended this throughout. The drug data refers to any encashed prescriptions in these BNF categories. We have changed the abstract main outcome measure to: "Encashed regular and as-required psychotropic prescriptions."

And added in the methods (pages 6-7):

"PIS is Scotland's electronic record of all encashed prescriptions (i.e. not prescriptions issued, nor drugs administered, but those that the carers/person with intellectual disabilities actually took to a pharmacist and exchanged for the drugs), and includes a record of the BNF code of each prescribed drug.(20) Prescribing information was then extracted, including both regular prescriptions and as-required medication."

And in the discussion (page 18):

"The period of 12 weeks extraction of PIS data was selected to account for the frequency of prescriptions being issued. It included both regular and as-required drugs; given the 12 week prescribing period it is likely that the as-required medication was being actively used (as a fresh prescription had been issued and was encashed by the person with intellectual disabilities/their carer during this period). The time period for encashment was identical at both time points for the linked cohort."

2. The method suggests that only prescriptions for a 12-week window were examined. It is unclear if regular vs. prn use could be determined.

***Response: We have rectified this omission by adding the following clarification (pages 6-7):

"Prescribing information was then extracted by BNF codes for the 3,906 adults with intellectual disabilities to identify all prescriptions of antipsychotics, antidepressants, antiepileptics, lithium, and hypnotics/anxiolytics across a 12-week prescribing window in 2014, including regular prescriptions and as-required medication."

And in the discussion (page 18):

"It included both regular and as-required drugs; given the 12 week prescribing period it is likely that the as-required medication was being actively used (as a fresh prescription had been given and was encashed by the person with intellectual disabilities/their carer during this period). The time period for encashment was identical at both time points for the linked cohort."

3. It should be acknowledged that the use of antidepressants (e.g. amitriptyline, duloxetine) and antiepileptics (e.g. pregabalin, gabapentin) for neuralgic pain has increased markedly over the timeframe examined (i.e. these drugs are often used for other conditions). Amitriptyline is a classic example that could have a multitude of uses in these patients (e.g. incontinence).

***Response: Yes, this is certainly the case for the general population. We think it is less relevant (unfortunately) for people with intellectual disabilities, who have difficulties communicating pain due to communication limitations or being non-verbal; and drug use for incontinence in this population would be highly atypical in Scotland. We have added the following to the limitations section of the discussion (page 18):

“Some antidepressants and antiepileptics have seen increased use in the general population over this time period for neuralgic pain. We do not know how relevant this is to people with intellectual disabilities who may have difficulties in communicating pain, and note that encashed antiepileptics did not increase between the two cohorts.”

4. Another limitation that should be included is the apparent lack of access to data on the individuals' other medical conditions (for which some of these drugs are commonly used...and more commonly than they were 10 years earlier).

***Response: We have detailed medical information at T1 for the T1 cohort and linked cohort, but not at T2. We included the statement below to address this possible limitation (page 18): “Some antidepressants and antiepileptics have seen increased use in the general population over this time period for neuralgic pain. We do not know how relevant this is to people with intellectual disabilities who may have difficulties in communicating pain, and note that encashed antiepileptics did not increase between the two cohorts.”

5. Please check the English of the second dot point under Strengths and limitations of the study.

***Response: Thank you for highlighting this – we have amended the sentence

6. Under Discussion: heading should state “Principal findings”

***Response: Thank you for highlighting this error which has now been amended

Reviewer 3:

Koji Kawakami, Kyoto University, Japan

1. It would be helpful if the authors explain why they selected the two points; 2002-2004 and 2014. Is there any specific reason such as the guideline was changed 2013 or something? Also, is there any reason why they compared for three years and just one year?

***Response: The selection of the time points was purely pragmatic. The T1 cohort was recruited in 2002-2004, so this time period was pre-defined. At T2, after several years of negotiation, circumstances were that we were able to gain ethical approval from 191/263 general practices for an extraction of their intellectual disabilities patient data. The decade time period allowed us to address our study aims. We have not added this explanation to the paper, but are happy to do so if the editor so advises.

The difference in the time period is due to the data collection methods – individual consenting and health assessments on all the 1,190 participants at T1 which took 2 years to complete, and electronic data extraction and linkage at T2. We have added the following to the methods so this information is more prominently made (page 6):

“Semi-structured individual health assessments, including medication review.....were conducted at T1..... Data collection was over a two year period.”

And (page 6-7):

“At T2, primary care records were record linked to Prescribing Information System (PIS) data,across a specific 12-week prescribing window in 2014.”

2. It would be clearer if the authors explain the aims of this study. This study consisted of two designs. It seems they compared the percentage of prescribed psychotropic drugs in the whole cohort of patients with intellectual disabilities and evaluated change in prescribed psychotropic drugs among patients who could be followed up for 10 years among whole cohort. Did the authors try to evaluate the prescriptions were followed the guideline? If so, does the guideline recommend to withdraw the medication? I mean, the long term use of drugs is not recommended?

***Response: The aim of the study is (page 5):

“The aim of this study is to investigate changes over a decade in psychotropic prescribing for adults with intellectual disabilities, and the diagnoses associated with antipsychotics, from detailed psychiatric assessments.” We appreciate the importance of this information, and therefore, so that this is more prominently made, we have introduced a sub-heading of “Aim” within the introduction.

Several guidelines recommend withdrawal of antipsychotics from people with intellectual disabilities who do not have psychosis, e.g. the National Institute for Health and Care Excellence guidelines on (1) mental health conditions for people with intellectual disabilities, and (2) behaviours that challenge in people with intellectual disabilities, and our study provides evidence that these guidelines are not being followed. We think the point made by the reviewer is important, and to reinforce this point (guidelines not being followed), we have added the references to the first sentence of the discussion (page 16).

3. Patients' selection in each period is different. Is it reasonable to compare with the two-periods of groups directly?

***Response: We believe the 2 comparisons made are valid, as (1) the prevalence of intellectual disabilities in the Health Board has not changed over this 10 year period, and (2) the linked cohort reports drugs by exactly the same participants at the two time points, 10 years apart. If the editor advises, we will add further information in the paper on this.

4. It would be more informative if the authors explain why they selected ‘multivariable repeated measures logistic regression models’ for analysis. Also, the reference should be stated here because the model is not popular.

***Response:

We believe that the phrase 'multivariable repeated measures logistic regression models' may have been misunderstood. We have replaced it with the following phrases, and additionally added that time was fitted as an "explanatory" factor (page 7)

"Each model included multiple explanatory variables, specifically; time as a binary variable to indicate each time point T1 and T2; sex; age as a continuous measure; level of intellectual disabilities as four-level categorical variable; presence of mental ill-health (yes/no, excluding problem behaviours); having problem behaviours (yes/no); and a binary dependant variable for each class of medication (yes/no).

5. What is the definition of 'change in prescription'. It should be stated in the statistical methods.

***Response: References to 'change in prescription' have been removed for clarity. We have amended the sentence that mentioned change in prescription to (page 7):

"This analysis was extended to explore whether there were associations between time or the subject characteristics at T1, with each prescription outcome using binary logistic regression models"

And have changed the title of table 6 to (page 14):

"Table 6: Multivariable analysis of exploratory T1 factors and time with psychotropic prescription for the linked cohort (N=545)"

6. As for the longitudinal design, the results have an immortal bias. They have to be followed for ten years. How did the authors consider if the patients quitted the treatment before ten years?

***Response:

We consider this an unavoidable bias in these types of studies, However we note that the characteristics of both the longitudinal and cross-sectional cohorts are broadly similar and therefore the results are generalisable .

7. As for the cross-sectional study, the prescriptions were not limited in newly prescribed drugs, right?

***Response: This is correct.

8. The authors mentioned, 'It appears that whilst people are not being withdrawn from antipsychotics once they commence them' in the discussion section. How we are able to know which prescriptions should be withdrawn. The patients might be needed the drugs use continuously.

***Response: This important point is related to the reviewer's previous comment, which we have therefore addressed in the same way, i.e: Several guidelines recommend withdrawal of antipsychotics from people with intellectual disabilities who do not have psychosis, e.g. the National Institute for Health and Care Excellence guidelines on (1) mental health conditions for people with intellectual disabilities, and (2) their guideline on behaviours that challenge in people with intellectual disabilities, and our study provides evidence that these guidelines are not being followed. We think the point made by the reviewer is important, and to reinforce this point (guidelines not being followed), we have added the references to the first sentence of the discussion (page 16).

9. In the abstract, the authors did not mention the result for longitudinal design. It should be stated. Also, the authors said 'rates of prescribing,' but it seems they did not consider the time such as person-year. Is 'rates' correct? I wonder it's instead 'percentage' or 'proportion.'

***Response: The abstract contains this sentence under the heading 'design':

"(b) Longitudinal cohort study with detailed health assessments at T1, and record linkage to T2 prescribing data."

And the second half of the results section of the abstract includes the longitudinal data as follow:

"(b) Psychotropics increased from 47.0% (256/545) in T1 to 57.8% (315/545) in T2 ($p < 0.001$): antipsychotics did not change (OR=1.18; CI (0.87, 1.60); $p = 0.280$), there was an increase for antidepressants (OR=2.80; CI 1.96, 4.00; $p < 0.001$), hypnotics/anxiolytics (OR=2.19; CI 1.34, 3.61; $p = 0.002$), and antiepileptics (OR=1.40; CI 1.06, 1.84; $p = 0.017$). Antipsychotic prescribing increased for people with problem behaviours in T1 (OR=6.45, CI 4.41, 9.45; $p < 0.001$), more so than for people with other mental ill-health in T1 (OR=4.11, CI 2.76, 6.11; $p < 0.001$)."

The individuals in the longitudinal cohort were recruited at approximately the same time, and followed up for approximately the same length of time; their medications are reported at the time of inception into the cohort and a fixed point of 12 weeks at follow up. We therefore did not consider person years to be a helpful addition.

10. Page 6, line 57. There is a typo: "reportedwith" to "reported with."

***Response: Thank you. This has been amended.

Reviewer 4

Ryan Smith Sultan, Columbia University, USA

1. Overall, writing is fairly casual and could benefit from a professional tone.

***Response: Changes to the document have been made throughout.

2. Overall, Needs proof reading, last paragraph of process and measures "complete data"

***Response: Thank you for pointing out this typographical error. The document has been fully proof-read prior to resubmission.

3. Overall, the article could benefit from a narrower focus. The article in some ways seem to be largely about antipsychotics—though the title says otherwise. Further, The data between T1 and T2 are different. Seems the article should be either about the comparison of the T 1 and T2 or looking at each individually. If it's the comparison, id remove the stuff about diagnoses and rates of meds as it can't be used in the second group (Table 3).

***Response: We understand the points the reviewer is making, but prefer to keep the focus as it is. Whilst we agree that antipsychotics are perhaps the most important drugs, the additional drug data contextualises prescribing in general in this population over this period. We have therefore added to the discussion (page 16): “The age-related change in antiepileptic prescribing in the linked cohort, but not in the comparison of the similarly aged whole cohorts, contextualises the antipsychotic and antidepressant findings (prescribing trends in general), as antiepileptics were almost all prescribed for the highly prevalent condition of epilepsy in this population.”

Additionally, the marked increase in antidepressants is notable and hence we believe it important to report.

The data between T1 and T2 are only different in that T1 includes people with intellectual disabilities across part of the Health Board whilst T2 includes people in all of the Health Board, and the method of drug data collection is unlikely to cause discrepancies in results.

Cross-sectional data is of interest (and we have necessarily reported it); we believe the longitudinal comparison of greater interest, particularly given the guideline emphasis in the UK of withdrawing antipsychotics, which our data shows is not being followed. If we did not report the mental ill-health data, it would be harder to interpret the extent of appropriateness of the prescriptions; hence, we prefer to retain this information in the report.

4. Article would benefit from a clinical lens. For example, clear evidence exists that antipsychotic medications are over used in this population. However, the article does not adequately discuss how individuals with intellectual disabilities often demonstrate severe issues and become dysregulated and often physically aggressive. Low dosing of antipsychotic medications often moderate these issues for ID individuals and allow them to stay in housing situations and avoid longer term institutionalization. This data set isn't able to examine that because these types of behavioral issues do not have a specific diagnostic code linked to them.

***Response: We do have more detailed information on the types of problem behaviours experienced by the individuals with intellectual disabilities at T1 and could add that – but – in the UK, clinical guidelines do not support use of antipsychotics for adults for the reasons the reviewer describes. Hence we have not commented further on this (other than quoting the guidelines), but would be happy to provide further information on the problem behaviours and associated drugs if the editor advises us to do so. In recognition that practice may differ in other countries, we have amended the first sentence of the discussion, to state the guidelines we quote are from the UK, as follows (page 16”): “Despite numerous calls and guidelines in the UK for the withdrawal of antipsychotic drugs from people with intellectual disabilities who do not have psychosis/bipolar disorders, our linked cohort analysis demonstrates no progress over a decade.”

[NB: Our research team includes two clinicians (a Consultant psychiatrist who specialises in intellectual disabilities, and a Consultant nurse who specialises in intellectual disabilities)].

5. Findings would benefit from a sub analysis that looked at level of intellectual disability (mild to profound) and likelihood of antipsychotic prescribing (and other psychotropics). I would imagine more profound ID is associated with higher prescribing and likely higher antipsychotic prescribing.

***Response: The logistic regression model reported in table 6 did include level of intellectual disabilities. The observed effects are described in the amended results section (page 12):

“Effects are also observed for level of intellectual disabilities. There was a gradient for antiepileptics (increased prescribing with increasing severity of intellectual disabilities) and a gradient for antidepressants (reduced prescribing with increasing severity of intellectual disabilities).”

6. Unsure of where the conclusion in the abstract about not withdrawing antipsychotics comes from. If that is a primary conclusion—it should be backed up by the results section of the abstract.

***Response: The data is provided on the linked cohort in the results section of the abstract: “(b) Psychotropics increased..... in T1 to..... in T2.....: antipsychotics did not change (OR=1.18; CI (0.87, 1.60); p=0.280).” We appreciate this is brief reporting, in view of the word count restriction in the abstract. We provide more information in the results section of the full text of the paper (table 5), and in the logistic regression models.

7. If the goal of this article was to link it to the concerns over use of meds in this population, particularly antipsychotics, I think the abstract needs to be more positive and less concerning sounding as the numbers clearly show they are reducing

***Response: We disagree; we have shown that antipsychotics are not being withdrawn as guidelines recommend, and think the abstract is balanced reporting.

8. Article would benefit from adding some citations from US data—which as examined this extensively, Some suggestions:
Sultan, R. S., Wang, S., Crystal, S., & Olfson, M. (2019). Antipsychotic Treatment Among Youths With Attention-Deficit/ Hyperactivity Disorder. *JAMA Network Open*, 2(7).
Sultan, R. S., Correll, C. U., Schoenbaum, M., King, M., Walkup, J. T., & Olfson, M. (2018). National Patterns of Commonly Prescribed Psychotropic Medications to Young People. *Journal of Child and Adolescent Psychopharmacology*, 28(3), 158–165.
<https://doi.org/10.1089/cap.2017.0077>
Olfson, M., King, M., & Schoenbaum, M. (2015). Treatment of young people with antipsychotic medications in the United States. *JAMA Psychiatry*, 72(9), 867–874.
<https://doi.org/10.1001/jamapsychiatry.2015.0500>

***Response: We have added this important research data on children/young people to the introduction (pages 3-4):

Antipsychotics are also frequently prescribed for children and young people with a range of developmental disabilities and problem behaviours (14, 15) and in the young general population, rates increase during adolescence. (16)”

Supplementary File
Please re-upload your supplementary file in PDF format.

***Response: Noted thank you.

VERSION 2 – REVIEW

REVIEWER	Gregory Peterson University of Tasmania, Australia.
REVIEW RETURNED	07-Jun-2020

GENERAL COMMENTS	Thank you for the revision. The queries have been addressed well. Some of the writing could still be improved e.g. bottom of page 8, is this correct: “At T1, 24.5% (292/1,190) of participants were prescribed antipsychotics and at T2 16.7% (653/3,906) were prescribed antidepressants. At T1, antidepressants were prescribed for 11.2% (133/1,190), and at T2 for 19.1% (746/3,906)..”?
---

VERSION 2 – AUTHOR RESPONSE

I have reviewed the paper and made revisions suggested by reviewer 2.

Yours sincerely,

Angela Henderson